# rpe v5: An emulator for reduced floating-point precision in large numerical simulations

Andrew Dawson[1] and Peter D. Düben[1]

[1]Atmospheric, Oceanic & Planetary Physics, Department of Physics, University of Oxford, Oxford UK

*Correspondence to:* Andrew Dawson (andrew.dawson@physics.ox.ac.uk)

**Abstract.** This paper describes the *rpe* library which has the capability to emulate the use of arbitrary reduced floating-point precision within large numerical models written in Fortran. The *rpe* software allows model developers to test how reduced floating-point precision affects the result of their simulations without having to make extensive code changes or port the model onto specialised hardware. The software can be used to identify parts of a program that are problematic for numerical precision and to guide changes to the program to allow a stronger reduction in precision.

The development of *rpe* was motivated by the strong demand for more computing power. If numerical precision can be reduced for an application under consideration while still achieving results of acceptable quality, computational cost can be reduced, since a reduction in numerical precision may allow an increase in performance or a reduction in power consumption. For simulations with weather and climate models, savings due to a reduction in precision could be reinvested to allow model simulations at higher spatial resolution or complexity, or to increase the number of ensemble members to improve predictions. *rpe* was developed with particular focus on the community of weather and climate modelling, but the software could be used with numerical simulations from other domains.

## 1 Introduction

Users of high performance computing (HPC) have historically enjoyed a steady increase in computing power according to Moore's Law, allowing the complexity of HPC applications to increase steadily over time. In recent years, advances in super-computing power have not come from increasingly fast processors, but from increasing the number of individual processors in a system. This has led to HPC applications being redesigned so that they scale well when run on many thousands of processors, and effort will be required to make sure models are efficient on the massively-parallel supercomputing architectures of the future. However, even for massively-parallel supercomputers a continuation of the exponential growth of computational power that has been observed in the past appears to be out of reach with current technologies, primarily since power consumption scales linearly with the number of processors such that power demand of future supercomputers will become excessive.

One approach to reducing the energy requirements of the next-generation supercomputer is to use so-called approximate computing or inexact hardware. This is hardware that makes some trade-off between accuracy and/or reproducibility, and speed and/or efficiency. A number of studies have shown the promising nature of such hardware implementations (see Hamilton et al., 2014; Xu et al., 2016, for an overview). One of the most promising avenues for inexact computing is the use of a reduction

in numerical precision. The precision of a floating-point number can be reduced by decreasing the number of bits used to represent the number. Storing and operating on fewer bits reduces the space in memory required for storage and could speed up computations on suitably designed hardware. Therefore using a reduced-precision representation for floating-point numbers could be a viable option for reducing the computational cost of complex numerical models, provided the results at reduced precision are acceptable.

The recognized standard for floating-point arithmetic is IEEE 754 floating-point (IEEE Computer Society, 2008). This standard specifies a format using a 32-bit representation for floating-point numbers (often referred to as single precision) and an additional option for a 64-bit double-precision representation. These representations are ubiquitous on modern consumer CPUs. It is possible to trade numerical precision for performance on existing hardware by switching from double to single precision. Using single-precision representations does not necessarily mean computation will be faster, often floating-point units are designed to perform optimally for 64-bit calculations, but for code that can be vectorized it is possible to process single precision at twice the speed as double precision. A key advantage of using a smaller representation is that more data can fit in memory and importantly in cache which can speed up memory-bound computations, and a smaller data volume is beneficial for parallel applications that communicate between processors via message passing. Significant savings in compute time are observed in many applications when precision is switched from double to single (e.g., Berger and Stamatakis, 2010; Rüdisühli et al., 2013; Vana et al., 2017). A representation using 16-bits has also been standardised upon by the IEEE, referred to as half precision. It is less common to find support for half precision on consumer hardware, although it is supported on some general purpose GPU hardware. Although consumer CPUs and GPUs typically offer only precision levels supported by IEEE 754, in principal hardware architectures are not limited to these three precision levels, and custom hardware (e.g., based on FPGAs) could be produced that uses an arbitrary number of bits to represent floating-point numbers.

Whilst aiming to understand the performance of complex numerical models on new hardware architectures with inexact processing capabilities it is useful to be able to experiment with different precision levels, including those that are not supported natively by currently available hardware. Programmable hardware could be used to study a reduction in precision for an application of interest (e.g., Düben et al., 2015a). However, this will require a total rewrite of the model to run on such hardware, which will often be a serious programming effort. For most applications it will be much easier to use software to emulate the behaviour of possible inexact hardware for a model of interest, thus avoiding the cost of re-engineering code for a new hardware platform without knowing what the results might be.

Tools for understanding rounding error in numerical simulations have been developed already. For example the CADNA library (Scott et al., 2007; Jézéquel and Chesneaux, 2008) can be used to estimate the contribution of round-off error in numerical calculations, and hence determine the likely accuracy of the solution. Along a similar line, the specialised compiler Verificarlo (Denis et al., 2016) is capable of estimating the number of real bits of precision in a calculation given the errors introduced by round-off errors at various "virtual" precisions. This approach has some problems when investigating rounding errors in weather and climate predictions. For example, for short term weather simulation an ensemble of forecasts is typically produced, and each member may have different small scale features such as the location of a particular thunderstorm, but all members are valid realisations of the state of the atmosphere. This forecasting methodology may allow us to predict if

thunderstorms are likely or not, but not the exact location of an individual storm. If the position of a particular storm is different in model using high precision and a model using lower precision, results can still be valid for both simulations (as with for different ensemble members) but the amount of local precipitation in each model can vary by orders of magnitude. This suggests that identifying the number of bits that are influenced by rounding errors may not be able to tell us what precision we need to make a useful prediction.

A variety of tools also exist for automatically tuning floating point precision, and producing mixed precision versions of double precision programs. Tools such as SAM (Graillat et al., 2011), Precimonious (Rubio-González et al., 2013), CRAFT (Lam et al., 2013) and PROMISE (Graillat et al., 2016) all provide automated methods for producing a correct mixed precision implementation. The automated tuning approach usually requires the specification of an appropriate error threshold used to determine the suitability of a particular automatically generated implementation of a program. This can be straightforward for problems where a numerical method is expected to converge to an exact solution, but becomes difficult for models geophysical flows. For these models there is typically considerable uncertainty in both the formulation of the model and the initial conditions. When we consider chaotic models, that when slightly perturbed will quickly diverge from each other (with the rate of divergence dependent on the weather regime of the initial conditions, and varying with time), it becomes difficult to specify such a condition. Two solutions may differ in the most significant bits of the results, yet still represent physically meaningful realisations of the model. In this situation it becomes necessary to perform explicit simulations, and for evaluation of the results to be performed by a domain scientist who understands the physical behaviour of the model, in order to determine the suitability of a particular reduced precision implementation.

This paper introduces the *rpe* (reduced precision emulator) software, which provides the capability to emulate calculations in arbitrary reduced floating-point precision, so that model developers may understand how reduced floating-point precision affects the result of their simulations. The *rpe* software allows modellers to study the impact of numerical precision in complex model simulations, while making only small changes to the model code. The software can also be used to identify parts of a program that are problematic for numerical precision and to guide changes to the program to allow a stronger reduction in precision. We refer to this as software co-design, and present an example of the co-design of a shallow water model in Sect. 3. *rpe* was developed with particular focus for the community of weather and climate modelling, but the software could be used with other numerical simulations from different domains to answer the same kind of questions.

Weather and climate models are becoming increasingly important in society. Even though weather and climate models are often run on some of the fastest computing hardware available, this is still not enough computational power to allow them to run at small scales that are believed to be important, for example resolving individual convective systems (Shukla et al., 2010; Prein et al., 2015). For weather and climate modelling, it was recently proposed that the use of reduced numerical precision may allow a significant reduction of computing cost (Palmer, 2014). Savings due to increased performance or reduced power consumption could be reinvested into larger computing systems and used to allow simulations at higher spatial resolution. To this end, a number of studies have already been performed on the use of reduced floating point precision in atmospheric applications that have used ancestors of the *rpe* tool that is presented in this paper (Düben et al., 2013, 2014; Düben and Palmer,

2014; Düben et al., 2015b, a; Thornes et al., 2017). Results show that numerical precision can be reduced significantly for the applications that have been considered.

Section 2 presents the *rpe* library including an overview on how to use *rpe*, how *rpe* works, how floating point truncation is realised within *rpe*, how to control numerical precision with *rpe*, and the basic software architecture. Section 3 presents results

for simulations with reduced numerical precision in a shallow water model that serves as an example for a possible application of *rpe*, and Section 4 provides the conclusions.

## 2 An emulator for reduced floating-point precision

In this section we introduce the *rpe* library and explain how to use the emulator, details on how the emulator truncates floating-point numbers, and how precision will propagate through a simulations. Details of the software architecture and possible pitfalls

and limitations of the emulator are also discussed.

### 2.1 How it works and how to use it

The *rpe* software is implemented as a Fortran library. This choice of language is due to the prevalence of Fortran in geophysical modelling. The emulator is implemented as a Fortran module that can be included in existing projects via the standard Fortran 90 modules mechanism. The module defines a new derived type named `rpe_var` (reduced precision emulator variable)

which can be substituted for the built-in Fortran `real` type in user code. The intention is to make the `rpe_var` type a drop-in replacement for the `real` type, allowing the introduction of reduced precision in complex numerical models to be manageable.

The emulator's core functionality comes from overloading the assignment (=) operator for `rpe_var` variables to ensure they are stored with a reduced precision. Overloaded versions of the remaining arithmetic and logic operators (+, -, *, /, **, ==, /=, >, <, >=, <=) that accept `rpe_var` instances as inputs are provided, along with overloaded versions of many of the

20 Fortran intrinsic functions that normally accept `real` types as input. Overloaded definitions that would normally return `real` types make use of the overloaded assignment internally to reduce the precision of their output appropriately (depending on the precision of the input) and return `rpe_var` types. Compound expressions are handled implicitly, with each intermediate operation involving `rpe_var` instances returning `rpe_var` instances of appropriate precision, before computing the next terms. Reduction of precision is applied after each operator or intrinsic function call. For intrinsics like `sin` we only reduce

the precision of the final result of the function application, even though the implementation of the intrinsic may internally consist of many individual floating-point operations. This provides a reasonable approximation to hardware reduced precision, while avoiding the complexity of reimplementing all numerical intrinsic functions from scratch.

Let us consider the program listed in Fig. 1a, which is implemented using single-precision floating-point. This program performs a simple numerical computation with a final value `P=118.081879`. This program could be rewritten to use a reduced

precision with 10 explicit significand bits by changing the type of the variables, as in Fig. 1b. The program using reduced precision computes the final value `P=118.0625`. The following operations are computed when this program executes:

**Algorithm 1** Operations performed by the reduced-precision program (Fig. 1b).

$$\texttt{rho} \leftarrow \mathcal{R}_{10}\left(1.2041\right)$$

$$\texttt{g} \leftarrow \mathcal{R}_{10}\left(9.80665\right)$$

$$\texttt{h} \leftarrow \mathcal{R}_{10}\left(10\right)$$

$$\texttt{P} \leftarrow \mathcal{R}_{10}\left(\texttt{h} \times \mathcal{R}_{10}\left(\texttt{rho} \times \texttt{g}\right)\right)$$

The notation $\mathcal{R}_n\left(\ldots\right)$ refers to an operator that reduces the precision of a floating-point value to $n$ significand bits and $\leftarrow$ indicates an assignment. The reduced precision is a consequence of using the `rpe_var` type instead of the `real` type. The reduced-precision type `rpe_var` has its own assignment operator that truncates values to the required number of significand bits before storing in memory, causing the values stored in the variables `rho`, `g`, and `h` to be truncated to 10 significand bits. The multiplication operator is also modified for `rpe_var` types, causing the expression for `P` to be evaluated by multiplying `rho` by `g`, truncating the result to 10 explicit significand bits, then multiplying this value by the value of `h` and once again truncating the result to 10 explicit significand bits. The assignment to `P` will also ensure truncation to 10 significand bits, but in this case it makes no change to the value stored, since it has already been truncated.

The reduced-precision program (Fig. 1b) sets a global default number of explicit significand bits for all `rpe_var` instances to 10 using `RPE_DEFAULT_SBITS=10`. It is also possible to control the precision for each variable independently as in the program listed in Fig 2, which produces the final value `P=118.0`. The operations performed by this program are listed below:

**Algorithm 2** Operations performed by the mixed precision program (Fig. 2).

$$\texttt{rho} \leftarrow \mathcal{R}_{8}\left(1.2041\right)$$

$$\texttt{g} \leftarrow \mathcal{R}_{9}\left(9.80665\right)$$

$$\texttt{h} \leftarrow \mathcal{R}_{10}\left(10\right)$$

$$\texttt{P} \leftarrow \mathcal{R}_{7}\left(\mathcal{R}_{10}\left(\texttt{h} \times \mathcal{R}_{9}\left(\texttt{rho} \times \texttt{g}\right)\right)\right)$$

This example demonstrates a more complex situation where multiple precisions are used, resulting in each multiplication casting to the precision of its most precise input (individual precision values are assigned using the `sbits` component). The behaviour of mixed-precision operations is described in detail in Sect. 2.3, including details of potential problems that may arise from the use of mixed precision.

## 2.2 Truncation of floating point numbers

Floating-point numbers consist of 3 parts: a sign bit that determines the sign of the number, an exponent that determines the magnitude of the number, and a significand that determines the digits precision of the number. The approach to reduced floating-point precision taken in this paper is a truncation of the significand. This truncation removes less significant bits from the floating-point significand, by setting them to zero, resulting in reduced precision. A naïve truncation that simply sets the least significant bits of the significand to zero would introduce a bias, since such a truncation is equivalent to always rounding

toward zero to the nearest representation with the desired number of remaining significand bits. We have encountered situations where such biases can have a significant impact on conservation properties when running simulations of computational fluid dynamics. Therefore it is necessary to apply rounding as well as truncation. The method used for reduction of precision in *rpe* emulates IEEE 754 rounding, where numbers are rounded to the nearest representation with the given number of bits, with the additional requirement that numbers exactly halfway between two representations are rounded to the representation with a 0 in the least significant bit. This scheme is unbiased, and thus avoids introducing bias errors as well as truncation errors. Figure 3 demonstrates the truncation process for the different rounding scenarios. The present implementation of our reduction of precision scheme only alters the significand (except where the exponent is changed due to rounding in accordance with IEEE 754 specifications), it does not currently implement a general emulation of exponent size. The exception to this is for IEEE 16-bit floating-point representations (half precision), where an optional strict emulation mode is provided. This option emulates half precision by adding extra checks to ensure the size of a truncated value is within the valid range for a 16-bit float with 5 exponent bits and 10 significand bits, and includes handling of subnormal values.

## 2.3 Controlling precision

An `rpe_var` instance is a structure that stores a floating-point value, along with an integer specifying the number of explicit significand bits the instance uses for storage. The *rpe* library supports mixed precision levels, allowing the precision of individual variables to be controlled independently, using `a%sbits=n` where `a` is an `rpe_var` and `n` is the number of bits that should be used. The number of explicit significand bits stored by an `rpe_var` instance can be set on a per-variable basis, or left unspecified in which case a default value will be used. The default number of significand bits is controlled by a module-level variable (`RPE_DEFAULT_SBITS`). This mechanism allows for different parts of a code to use different precision levels, and provides the ability to change the precision level used without recompiling.

When an operation is performed on an `rpe_var` instance the number of significand bits the instance has is looked up and used to determine the precision required in the resulting value. For operations on values with different precisions, the output has the same number of significand bits as the input with the largest number of significand bits (i.e., the input with the highest precision). This behaviour was chosen to mimic the familiar behaviour when performing a binary operation with a 32-bit and a 64-bit floating-point number, where the 32-bit number is promoted to a 64-bit number and the result is a 64-bit floating-point number. For example, multiplying an `rpe_var` instance that has 10 significand bits by an `rpe_var` instance that has 12 significand bits will produce a value that has 12 significand bits. This analogous behaviour is extended to operations that mix `rpe_var` instances and intrinsic types. For the purposes of determining the precision of the result of such an operation 32- and 64-bit `real` instances are considered to have their actual number of explicit significand bits (23 and 52 respectively), while other compatible types are deemed to have zero significand bits. This choice ensures that `rpe_var` types interact with `real` types in a way that is predictable and consistent with how mixing intrinsic `real` types with different precision behaves, whilst ensuring that performing an operation involving an `rpe_var` and an integer will preserve the precision of the `rpe_var` instance. For example, multiplying a `rpe_var` instance with 10 significand bits by a 32-bit real type will produce

an `rpe_var` instance with a value truncated to 23 significand bits (the number in a single-precision float), but multiplying the same `rpe_var` instance by an integer will yield an `rpe_var` instance still having only 10-bits in its significand.

Because compound expressions are evaluated as separate intermediate terms care must be taken to ensure computations are expressed at the required precision. For example, consider the computation `b * (a + 2.0)`, where the variables `a` and `b` are instances of the type `rpe_var` with 10 significand bits. The literal floating point value `2.0` is treated as a single-precision floating-point number, and therefore the result of the expression `a + 2.0` has 23 bits of precision in the significand (single precision), which in turn carries more precision than is desired into the multiplication with `b`, the result of which will also have 23 bits of precision in the significand. This type of computation can be described as "leaking" higher precision into the expression through implicit casting. The *rpe* library provides a convenient way to resolve this for the case of literal numeric values, the previous computation could be written as `b * (a + rpe_literal(2.0, 10))` which converts the value `2.0` to an `rpe_var` type instance with 10 significand bits and thus the intermediate term `a + 2.0` would also be evaluated at 10 significand bit precision. This example uses floating-point literals, since this is a common 'gotcha' when implementing code with *rpe*, but the same problem can occur when performing compound operations on `rpe_var` instances that have different precision levels, and thus this issue should always be considered when utilising mixed precision.

It is possible to change precision at runtime, either by changing the module-level variable `RPE_DEFAULT_SBITS` or by changing the `sbits` component of an `rpe_var` instance. However, doing so is likely to introduce subtle problems. The main issue with this is that changing the number of bits in the significand of a variable to a smaller number of bits does not truncate the value of the variable to the new number of significand bits automatically. This problem can be mitigated by manually truncating variables whose precision has changed, although this may not be straightforward if precision is changed by setting `RPE_DEFAULT_SBITS`. Therefore we generally discourage users from changing precision at run time.

The result of truncating double-precision values to 23 significand bits (the number used by IEEE single precision) with the emulator is the same as casting from double to single precision using built-in types. The emulator internally uses a 64-bit `real` type to store all values, which contains 11 exponent bits rather than the 8 used in single precision, and therefore it is possible to store values in almost the full range of double precision using a truncated significand.

## 2.4 Software architecture

The emulator software consists of a relatively small hand-coded core. This core implements the `rpe_var` derived type, functions and subroutines implementing the algorithms for reduction of floating-point precision, and the overloading of assignment operators. The rest of the library, including the overloaded operator and intrinsic function definitions, is computer generated using a code generator supplied as part of the software. Using a code generator to produce the Fortran code for overloaded definitions has a number of advantages. Firstly it allows a developer to introduce changes to the way the overloaded definitions are implemented without having to modify each individual definition by hand. This is beneficial from the point of view of software quality, as it is less likely that errors will be made resulting in the reduction of precision being handled differently by different operators. Secondly this design makes extending the set of overloaded intrinsic functions simpler to manage and maintain. Adding a new overloaded definition requires adding a short definition to a configuration file in JSON format (a

human-readable text format; ECMA, 2013) that the code generator uses to write the necessary Fortran code to implement it, something which could be done easily by someone not familiar with the structure of the emulator code.

The software is provided with a suite of tests to verify its functionality. These tests are used for development, and can be run by the end user to verify their build of the software is functioning as expected. The tests are divided into two categories: unit tests and integration tests. The unit tests test the functionality of individual components of the emulator, isolated from other components. The integration tests verify the performance of the emulator by running a simple Lorenz attractor model. These tests also serve as a useful example on how to use the emulator.

## 3 Example application: A shallow-water model with reduced numerical precision

In this section, we present a precision analysis for a shallow water model, a typical medium-sized model setup that is often studied in the development of atmosphere and ocean models, in a Munk double-gyre configuration. The model under investigation is a standard approach to implement the shallow water equations on an Arakawa-C grid (Arakawa and Lamb, 1977), based on the model used in Marshall et al. (2013). The same model has been studied in detail on FPGA hardware (Targett et al., 2015). However, this required the model to be completely re-written to run on the specific hardware. We use the *rpe* library to make simple tests to determine if precision can be reduced beyond single precision, and how the model can be modified to make it more resilient at lower precisions.

The prognostic parameters in the shallow water model follow the equations:

$$\partial_t u - \left(f + \bar{\zeta}^y\right)\bar{v}^{xy} + \partial_x B = \nu\nabla^2 u + \tau_x, \tag{1}$$

$$\partial_t v - \left(f + \bar{\zeta}^x\right)\bar{u}^{xy} + \partial_y B = \nu\nabla^2 v + \tau_y, \tag{2}$$

$$\partial_t h + \partial_x\left((h_0 + \bar{h}^x)u\right) + \partial_y\left((h_0 + \bar{h}^y)v\right) = 0, \tag{3}$$

where $B = gh + \frac{1}{2}\left(\bar{u}^x\right)^2 + \frac{1}{2}\left(\bar{v}^y\right)^2$ is the Bernoulli potential, $\zeta = \partial_x v - \partial_y u$ is relative vorticity, $u$ and $v$ are the zonal and meridional components of the velocity, $f$ is the Coriolis parameter, $g$ is the gravitational acceleration, $h_0$ is the mean fluid depth, $h$ is the surface elevation, and $\tau_x$ and $\tau_y$ represent the wind forcing in zonal and meridional direction. The over-bars indicate averaging in $x$ or $y$ direction on the C-grid to map field values between edges and cell centres. The Munk double-gyre test case is a standard model setup in oceanography, and the specific configuration for this paper is based on model simulations in Cooper and Zanna (2015). The domain size is $3,480 \times 3,480$ km, resolved on a grid with $100 \times 100$ points. We use a mean fluid depth $h_0 = 500$ m, a viscosity of $\nu = 470.23$ m$^2$ s$^{-1}$, the Coriolis parameter describes a beta plane ($f = f_0 + \beta y$ with $f_0 = 4.46 \times 10^{-5}$ s$^{-1}$ and $\beta = 2.0 \times 10^{-11}$ m$^{-1}$ s$^{-1}$). The wind forcing is zero in the $y$-direction ($\tau_y = 0$) and $\tau_x$ ranges from $-0.4$ to $0.2$ Pa, varying with latitude in a sinusoidal structure to force the double gyre. A third order Adams-Bashforth time-stepping scheme with a time-step of $25$ s is used for time integration.

A double-precision experiment acts as a reference, we are aiming to reproduce this reference as closely as possible using reduced precision. A first-order estimation of the effect of reduced precision is achieved by running a single-precision simulation using native 32-bit representation (although a version emulating 23-bit precision has also been run, and is of similar quality).

Two runs with reduced precision are produced using *rpe*, one with 15 bits of precision in the floating-point significand, and one with 10 bits (the number of significand bits in an IEEE half-precision float).

Each simulation is spun-up for 5 million time-steps to reach equilibrium, then integrated for a further 15 million time-steps. Figure 4 shows a snapshot of the model simulations for each of the experiments some time after spin-up. The simulation using single precision (Fig. 4b) is qualitatively similar to the reference double-precision simulation (Fig. 4a) in that it shows height variations with similar magnitudes, with no suggestion of numerical stability problems. These snapshots are not expected to be identical between the different precision configurations due to the chaotic nature of the simulations, but similar flow behaviour would be expected. The simulation with emulated 15-bit significands (Fig. 4c) also shows smooth fields with height features of similar magnitude as in the double-precision reference simulation. The surface height field in the simulation with 10-bit significands (Fig. 4d) has smaller magnitude variations than the reference simulation, and some small-scale noise features can be seen as jagged contours, particularly noticeable in the lower part of the domain. These features suggest the dynamics of the simulation with 10-bit significands are being perturbed by a lack of precision. The problem in the 10-bit experiment is that the increment of the time-step is typically much smaller than the magnitude of the prognostic variable. In general, adding a small floating-point number to a large floating-point number can result in a loss of precision. We can avoid this error by maintaining a higher-precision prognostic variable, while computing the expensive right-hand-side tendency with emulated half precision. This scheme is illustrated in Algorithm 3. In this co-designed 10-bit experimental configuration, all floating-point numbers are half precision except for the prognostic fields and a few parameters used for model initialisation (which don't affect the running cost). A snapshot of this experiment is shown in Fig. 4e. Unlike the original 10-bit experiment, the co-designed 10-bit experiment has a smooth height field showing no indications of numerical stability problems.

---

**Algorithm 3** Time-stepping of a prognostic variable with mixed precision, using a simple forward in time scheme.

---

    **while** time-stepping **do**
        $P_{\text{low}} \leftarrow P_{\text{high}}$
        $x \leftarrow \text{COMPUTERHS}\left(P_{\text{low}}\right)$
        $P_{\text{high}} \leftarrow P_{\text{high}} + \Delta t \times x$
    **end while**

---

Figure 5 shows the mean height field from the reference double-precision simulation, the 15-bit significand simulation, and the co-designed half-precision simulation. The mean height field is qualitatively similar in all 3, suggesting a reduction in floating-point precision is acceptable for this model. We have used the *rpe* software to determine that the shallow water model can accept a strong reduction in numerical precision, and thus trading numerical precision against performance appears to be a reasonable approach for this model. In practice we could now continue to consider implementations of this model on real reduced-precision hardware, such as GPUs supporting 16-bit floating-point arithmetic, or an application specific hardware implementation.

The shallow water model provides a good context to discuss the effect of rounding on conservation properties. The surface height integrated over the model domain is conserved in the shallow water equations. However in a numerical approximation

we expect that there will be some error introduced due to truncation. For the reference double precision simulation the mean surface height of the fluid changes by $\sim 10^{-17}$ m over 20 million time-steps, in the single precision simulation the change is $\sim 10^{-7}$ m, and in the co-designed half precision simulation it is $\sim 10^{-3}$ m. Although the height error at lower precision is increased greatly over the double precision simulation, the absolute amount of fluid lost is small compared to the domain size. If this is a concern then a mass-fixing scheme could be introduced to mitigate. Perhaps more interesting is the influence of the rounding scheme used when truncating floating-point significands. If we truncate without rounding (equivalent to always rounding towards zero), a large bias is introduced into the simulation, to the point where it will fail to complete due to numerical errors. If instead we choose to round to the nearest representation (as IEEE but without the special case for numbers half way between two representations illustrated in Fig. 3d) we see a height error of $\sim 10^{-2}$ m. The difference between the round-to-nearest and the round-to-nearest with tie-to-even rounding schemes is not negligible, highlighting the value of unbiased rounding on computing conserved properties.

## 4   Discussion and Conclusions

A tool for running numerical simulations at emulated reduced floating-point precision has been described and documented. The efficacy of this tool has been demonstrated on a simplified geophysical model, and in particular the value of using *rpe* to co-design a model for lower precision is demonstrated.

There are a few notable limitations of the emulator, which may affect how it can be applied. The lack of support for emulating different exponent sizes may be relevant to certain applications. In principle it would be possible to extend the software to support emulation of arbitrary exponent sizes below the 11-bit significand of IEEE double precision. This could be done by enforcing a check on the size of each truncated value as it is produced by the truncation procedure. However, these checks will create additional computational overheads.

It is also worth noting that the solution obtained with the emulator is unlikely to be bit-reproducible compared to a solution obtained using native hardware support for a given significand size. There are several factors contributing to this, such as the treatment of intrinsics as atomic operations when they may actually be composed of multiple floating-point operations and the ability of compilers to optimize code using the emulator. The emulator operates at the Fortran source level, before compiler optimizations, and therefore code compiled with the emulator is likely to be optimized differently than code not using native reduced precision. For example, the compiler may take advantage of a fused-multiply-add (FMA) operation that performs a multiply and an add operation in one step with a single rounding at the end, as opposed to rounding an intermediate product to a given number of significand digits before computing the addition and rounding the final result. Such an optimisation is unlikely to be available when using the `rpe_var` type due to the complexity of the overloaded multiply and add operators. However, bit-reproducibility can also not be guaranteed between simulations that use IEEE double or single precision on different computing hardware or even different compiling flags, so this may not be a primary concern.

Using the emulator for reduced-precision simulations will introduce a significant overhead for all floating-point operations, and will result in poorer performance. While the emulator is designed to be used to study a possible precision reduction to

allow simulations that run faster on real hardware, it is important to realise that the focus of the software is being able to verify the output of a model at reduced precision rather than to emulate the actual efficiency gains one might expect from reduced precision. The overhead introduced by the emulator may vary depending on its application. In our experience the overhead of the library in terms of execution time is at least a factor of 10. Tests on the simple chaotic dynamical system of Lorenz (1963) suggest an execution time penalty of approximately 30 times, compared to a double precision simulation, and the co-designed shallow water model detailed in Sect. 3 experiences an execution time penalty of approximately 70 times compared to the double precision implementation, both tested using *gfortran* 4.8 with optimization level 2 to compile both the models and *rpe* itself.

The *rpe* software can, in principle, be used as a basis to emulate other hardware setups, such as hardware that shows bit flips occasionally. For most hardware setups, a simple replacement of the function that is actually realising the bit truncation will be sufficient. It can also be anticipated that it is possible to change the emulator to mimic the behaviour of specific error patterns of specific reduced-precision hardware (e.g., Düben et al., 2014).

## 5 Code availability

The *rpe* software is freely available under the Apache 2.0 license and may be accessed at http://github.com/aopp-pred/rpe. The repository contains the source code for the Fortran emulator, the code generator, the unit and integration test suites, and the source code for the documentation. Rendered documentation for recent versions is available to view at http://rpe.readthedocs. io. The code for the shallow water model configurations discussed in Sect. 3 are available at http://github.com/aopp-pred/ rpe-examples.

## 6 Author contribution

A. Dawson and P. Düben developed the *rpe* software and prepared the manuscript. P. Düben developed and carried out the shallow water model experiments.

*Acknowledgements.* The authors thank Mike Giles for useful discussion of IEEE rounding and code review, and Tim Palmer and Sam Hatfield for useful feedback and motivation. The authors received funding from ERC grant 291406 "Towards the Prototype Probabilistic Earth-System Model for Climate Prediction".

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

4     g = 9.80665
5     h = 10
6     P = rho * g * h
7     write (*, *) P
8 end program
```

**(b)**

```
1 program reduced_precision
2     use rp_emulator
3     type(rpe_var) :: rho, g, h, P
4     RPE_DEFAULT_SBITS = 10
5     rho = 1.2041
6     g = 9.80665
7     h = 10
8     P = rho * g * h
9     write (*, *) P%val
10 end program
```

**Figure 1.** A simple numerical program using (a) standard single-precision arithmetic, and (b) reduced-precision arithmetic with 10 significand bits.

```
1 program mixed_precision
2     use rp_emulator
3     type(rpe_var) :: rho, g, h, P
4     rho%sbits = 8
5     g%sbits = 9
6     h%sbits = 10
7     P%sbits = 7
8     rho = 1.2041
9     g = 9.80665
10     h = 10
11     P = rho * g * h
12     write (*, *) P%val
13 end program
```

**Figure 2.** A simple numerical program utilising emulated mixed precision arithmetic.

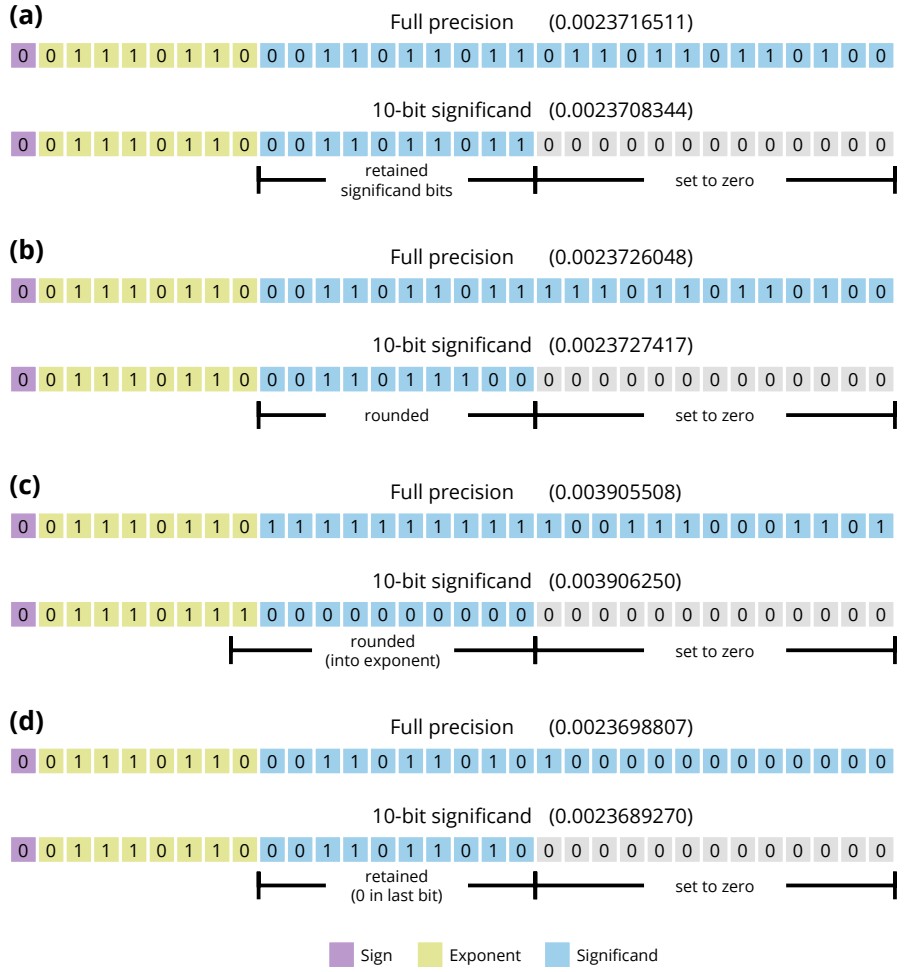

**Figure 3.** Examples of different rounding scenarios when truncating 32-bit floating-point numbers to 10 significand bits. (a) Bit 13 (counted from the right; the left-most bit being truncated) is a zero, therefore no rounding is necessary. (b) Bit 13 is a 1 which triggers rounding. (c) Bit 13 is a 1 which triggers rounding, in this case the rounding reaches the exponent, which is permitted by IEEE 754. (d) The number is exactly halfway between two 10-bit significand representations, this is a special case in IEEE 754, and always results in rounding to the representation with a 0 in the least significant bit.

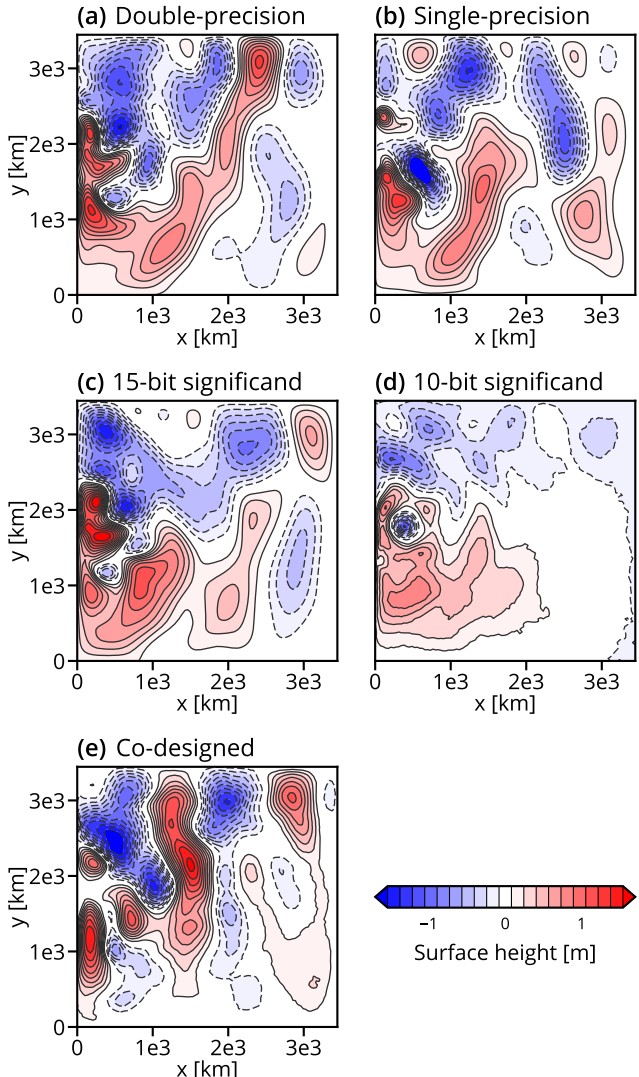

**Figure 4.** Snapshot of the surface height field for simulations of the shallow water model. (a) Double-precision reference simulation, (b) single-precision simulation, (c) reduced precision with 15-bit significand for all floating-point numbers, (d) reduced precision with 10-bit significand for all floating-point numbers, and (e) simulation co-designed for half precision.

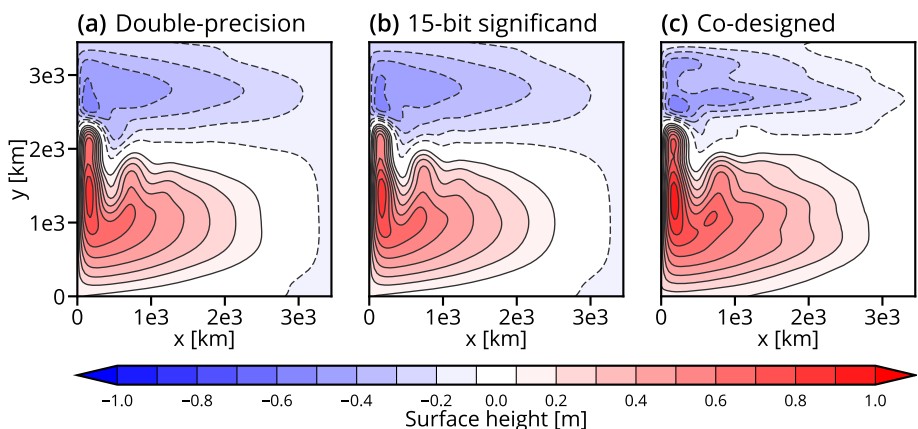

**Figure 5.** Time-mean surface height field for simulations of the shallow water model calculated over 15 million time steps. (a) Double-precision reference simulation, (b) reduced precision with 15-bit significand for all floating-point numbers, and (c) simulation co-designed for half precision.