# Peer review of "rpe v5: An emulator for reduced floating-point precision in large numerical simulations"

_Geoscientific Model Development, 2016_

## Referee Comment (RC2) · Anonymous Referee #2 · 23 Feb 2017

General comments:

This article presents a library (rpe v5) that enables one to change the precision of floating-point variables in Fortran codes. Reducing the precision can improve the code performance and it can be particularly advantageous in a parallel environment because it enables one to reduce the volume of data exchanged. This article recalls the importance of mixed precision simulations. The rpe v5 library seems easy to use in Fortran codes, the Fortran language being widespread for geoscience simulations. However the interest of the library seems limited compared to existing tools. As remarked by the authors, emulated reduced precision may produce different results than the native one. In rpe v5, with reduced precision variables, the exponent range remains the same, except when half-precision variables are emulated. No information is given on the validity of the results produced. Users must perform comparisons with, for instance, the double

precision results that are supposed to be correct.

Specific comments:

In the introduction two tools that estimate rounding errors are mentioned: CADNA and Verificarlo. In the reference "Scott et al." an author is missing (C. Denis). However another article describes CADNA in more detail: F. Jézéquel, J.-M. Chesneaux, CADNA: a library for estimating round-off error propagation, Computer Physics Communications, 178(12), pages 933-955, 2008.

In my opinion, a state of the art relative to precision reduction is missing.

Introducing errors on floating-point variables in a Fortran code is a functionality of CADNA. The SAM (Stochastic Arithmetic in Multiprecision) library enables one to control rounding errors in arbitrary precision simulations. Furthermore tools exist to auto-tune floating-point precision; some are listed below.

CRAFT HPC M.O. Lam, J.K. Hollingsworth, B.R. de Supinski, M.P. Legendre, Automatically Adapting Programs for Mixed-precision Floating-point Computation, in: Proceedings of the 27th International ACM Conference on International Conference on Supercomputing, ICS '13, ACM, New York, NY, USA, 2013, pp. 369–378. https://sourceforge.net/projects/crafthpc/

Precimonious C. Rubio-González, C. Nguyen, H.D. Nguyen, J. Demmel, W. Kahan, K. Sen, D.H. Bailey, C. Iancu, D. Hough, Precimonious: Tuning assistant for floating-point precision, in: Proceedings of the International Conference on High Performance Computing, Networking, Storage and Analysis, SC'13, ACM, New York, NY, USA, 2013, pp. 27:1–27:12. https://github.com/corvette-berkeley/precimonious

Blame Analysis C. Nguyen, C. Rubio-González, B. Mehne, K. Sen, J. Demmel, W. Kahan, C. Iancu, W. Lavrijsen, D.H. Bailey, D. Hough, Floating-Point Precision Tuning Using Blame Analysis, Technical Report, LBNL TR, 2015. https://github.com/corvette-berkeley/shadow-execution

PROMISE S. Graillat, F. Jézéquel, R. Picot, F. Févotte, B. Lathuilière. PROMISE: floating-point precision tuning with stochastic arithmetic, 17th international symposium on Scientific Computing, Computer Arithmetic and Validated Numerics (SCAN 2016), Uppsala (Sweden), September 2016. http://promise.lip6.fr/

Section 2.2: It would be interesting to have more details on the impact of rounding toward zero on fluid dynamics simulations. What would be the impact of rounding to plus or minus infinity on conservation properties?

As a remark, being not familiar with fluid dynamics simulations, I cannot easily juge the similarity of results displayed in Figure 4. The similarity of results displayed in Figure 5 seems clearer. However it would be interesting to compare them with those obtained with 10-bit significand.

The performance overhead due to the library is mentioned in the conclusion. It would be interesting to have measurements of that overhead.

Technical corrections are proposed below.

p4: The reduced-precision program (Fig. 1b) set ->The reduced-precision program (Fig. 1b) sets

p7: Therefore we generally we discourage -> Therefore we generally discourage

---

## Author Comment (AC1) · 13 Mar 2017

We thank the reviewer for their feedback and we are happy to address the comments in a revised version of the paper, and discuss the reviewer's comments in a detailed reply. In particular, we are happy to add:

- A more discussion on other tools that investigate the propagation of rounding errors through numerical integrations such as the tools named in the review.

- More details on the impact of rounding to plus or minus infinity on conservation properties.

- A more intuitive discussion of the results in Figure 4.

- More details and discussion on the performance overhead, including measurements.

However, we would like to stress already in this very short reply that rpe was designed particularly for fluid dynamical simulations that show chaotic dynamics in a complex model setup. Tools such as CADNA, Precimonious etc. work well for problems where there is a precise exact solution to be obtained. However, they are not as useful to investigate models for geophysical flows. For these models there is typically considerable uncertainty in both the formulation of the model and the initial conditions. Due to the chaotic nature model simulations that are only slightly perturbed, for example by the use of lower or higher precision, will quickly diverge from each other (with the rate of divergence proportional to the leading Lyapunov exponent that will depend on the weather regime of the initial conditions and vary with time). To perform reliable forecasts we need to perform ensembles of simulations that can show good predictive skill even if rounding errors have reached the most significant digits for some of the ensemble members and model outputs. Tools such as CADNA fail to provide satisfying results in a chaotic regime for which a rounding errors in one direction can result in a response of the system in the exact opposite direction. It was these intricacies of analysing precision in chaotic geophysical models that motivated the development of *rpe*.

It is true that *rpe* will not provide information on the validity of results and that the user will need to compare against double precision results. However, this is intentional since a quantification of rounding errors does not provide the required insight in our model framework. Take the paper *Düben et al.* (2017) as an example, where a study of reduced precision arithmetic for a cloud resolving model coupled to a global model is made. To identify model quality by looking at the impact of rounding errors on the output parameters of the cloud resolving model would not provide sufficient insight; model results may differ up to the most significant digit for some of the output parameters of the cloud resolving model due to exponential growth of error in the initial conditions.

However, the cloud resolving model at reduced precision may still provide physically meaningful results that provide sufficient feedback to the global model simulations. This can only be tested with explicit simulations using a tool such as *rpe*, and evaluation by a domain scientist who understands the physical behaviour of the model.

**References**

Düben, P. D., A. Subramanian, A. Dawson, T. N. Palmer (2017), A study of reduced numerical precision to make superparameterization more competitive using a hardware emulator in the OpenIFS model, *J. Adv. Model. Earth Syst.*, doi: 10.1002/2016MS000862.

---

## Author Response (AR1)

**Author's response: rpe v5: An emulator for reduced floating-point precision in large numerical simulations**

Andrew Dawson[1] and Peter D. Düben[1]

[1]Atmospheric, Oceanic & Planetary Physics, Department of Physics, University of Oxford, Oxford UK

*Correspondence to:* Andrew Dawson (andrew.dawson@physics.ox.ac.uk)

**1 Comments from referees**

We thanks the reviewers for their helpful comments. We have responded point-by-point. Comments from the reviewer are in blue text, and our responses are in black. Verbatim text from the revised manuscript is indicated by *italics*. Where additions or modifications have been made to the text, we provide the page and line number as well as the revised text.

**1.1 Anonymous reviewer #1**

N/A

**1.2 Anonymous reviewer #2**

This article presents a library (rpe v5) that enables one to change the precision of floating-point variables in Fortran codes. Reducing the precision can improve the code performance and it can be particularly advantageous in a parallel environment because it enables one to reduce the volume of data exchanged. This article recalls the importance of mixed precision simulations. The rpe v5 library seems easy to use in Fortran codes, the Fortran language being widespread for geoscience simulations. However the interest of the library seems limited compared to existing tools. As remarked by the authors, emulated reduced precision may produce different results than the native one. In rpe v5, with reduced precision variables, the exponent range remains the same, except when half-precision variables are emulated. No information is given on the validity of the results produced. Users must perform comparisons with, for instance, the double precision results that are supposed to be correct.

> The motivation behind the development of *rpe* was to enable the execution of manually chosen reduced (mixed) precision simulations in geophysical models. It is true that *rpe* will not provide information on the validity of results and that the user will need to compare against double precision results. However, this is intentional since a quantification of rounding errors does not necessarily provide the required insight in our model framework. The intended audience for the software is users of geophysical models that in particular feature complex chaotic dynamics and large inherent uncertainties in both formulation and initial conditions, which present a problem for automatic floating-point precision tuning. We hope our responses to the specific comments below, and the revisions to the manuscript, will make this more clear.

**1.2.1 Specific comments:**

In the introduction two tools that estimate rounding errors are mentioned: CADNA and Verificarlo. In the reference "Scott et al." an author is missing (C. Denis). However another article describes CADNA in more detail: F. Jézéquel, J.-M. Chesneaux, CADNA: a library for estimating round-off error propagation, Computer Physics Communications, 178(12), pages 933-955, 2008.

> We have added the missing author of Scott et al. (2007) to the reference list. We have also added a reference to Jézéquel and Chesneaux (2008) in the text (page: 2; line: 29).

In my opinion, a state of the art relative to precision reduction is missing. Introducing errors on floating-point variables in a Fortran code is a functionality of CADNA. The SAM (Stochastic Arithmetic in Multiprecision) library enables one to control rounding errors in arbitrary precision simulations. Furthermore tools exist to autotune floating-point precision; some are listed below.

– CRAFT HPC M.O. Lam, J.K. Hollingsworth, B.R. de Supinski, M.P. Legendre, Automatically Adapting Programs for Mixed-precision Floating-point Computation, in: Proceedings of the 27th International ACM Conference on International Conference on Supercomputing, ICS '13, ACM, New York, NY, USA, 2013, pp. 369–378.

  https://sourceforge.net/projects/crafthpc/

– Precimonious C. Rubio-González, C. Nguyen, H.D. Nguyen, J. Demmel, W. Kahan, K. Sen, D.H. Bailey, C. Iancu, D. Hough, Precimonious: Tuning assistant for floating-point precision, in: Proceedings of the International Conference on High Performance Computing, Networking, Storage and Analysis, SC'13, ACM, New York, NY, USA, 2013, pp. 27:1–27:12. https://github.com/corvette-berkeley/precimonious

– Blame Analysis C. Nguyen, C. Rubio-González, B. Mehne, K. Sen, J. Demmel, W. Kahan, C. Iancu, W. Lavrijsen, D.H. Bailey, D. Hough, Floating-Point Precision Tuning Using Blame Analysis, Technical Report, LBNL TR, 2015. https://github.com/corvetteberkeley/shadow-execution

– PROMISE S. Graillat, F. Jézéquel, R. Picot, F. Févotte, B. Lathuilière. PROMISE: floating-point precision tuning with stochastic arithmetic, 17th international symposium on Scientific Computing, Computer Arithmetic and Validated Numerics (SCAN 2016), Uppsala (Sweden), September 2016. http://promise.lip6.fr/

> We would like to stress that *rpe* was designed particularly for fluid dynamical simulations that show chaotic dynamics in a complex model setup. This scenario is somewhat different from what the existing tools are designed to address. However, we acknowledge that our discussion of the state-of-the-art was incomplete. We have modified our introduction section to include the following paragraphs, including references to a wider number of works in this area, and also describing in more detail our motivation for developing *rpe* (page: 2; line: 28):

*"Tools for understanding rounding error in numerical simulations have been developed already. For example the CADNA library (Scott et al., 2007; Jézéquel and Chesneaux, 2008) can be used to estimate the contribution of round-off error in numerical calculations, and hence determine the likely accuracy of the solution. Along a similar line, the specialised compiler Verificarlo (Denis et al., 2016) is capable of estimating the number of real bits of precision in a calculation given the errors introduced by round-off errors at various "virtual" precisions. This approach has some problems when investigating rounding errors in weather and climate predictions. For example, for short term weather simulation an ensemble of forecasts is typically produced, and each member may have different small scale features such as the location of a particular thunderstorm, but all members are valid realisations of the state of the atmosphere. This forecasting methodology may allow us to predict whether thunderstorms are likely or not, but not the exact location of an individual storm. If the position of a particular storm is different in model using high precision and a model using lower precision, results can still be valid for both simulations (as with for different ensemble members) but the amount of local precipitation in each model can vary by orders of magnitude. This suggests that identifying the number of bits that are influenced by rounding errors may not be able to tell us what precision we need to make a useful prediction.*

*A variety of tools also exist for automatically tuning floating point precision, and producing mixed precision versions of double precision programs. Tools such as SAM (Graillat et al., 2011), Precimonious (Rubio-González et al., 2013), CRAFT (Lam et al., 2013) and PROMISE (Graillat et al., 2016) all provide automated methods for producing a correct mixed precision implementation. The automated tuning approach usually requires the specification of an appropriate error threshold used to determine the suitability of a particular automatically generated implementation of a program. This can be straightforward for problems where a numerical method is expected to converge to an exact solution, but becomes difficult for models geophysical flows. For these models there is typically considerable uncertainty in both the formulation of the model and the initial conditions. When we consider chaotic models, that when slightly perturbed will quickly diverge from each other (with the rate of divergence dependent on the weather regime of the initial conditions, and varying with time), it becomes difficult to specify such a condition. Two solutions may differ in the most significant bits of the results, yet still represent physically meaningful realisations of the model. In this situation it becomes necessary to perform explicit simulations, and for evaluation of the results to be performed by a domain scientist who understands the physical behaviour of the model, in order to determine the suitability of a particular reduced precision implementation."*

Section 2.2: It would be interesting to have more details on the impact of rounding toward zero on fluid dynamics simulations. What would be the impact of rounding to plus or minus infinity on conservation properties?

We have performed simulations of the shallow water model described in the paper, using various implementations of rounding. Using the full IEEE 754 rounding scheme (round-to-nearest, tie-to-even) we see an approximately conserved surface height field after 20 million time-steps. We have also used a simple explicit rounding scheme,

where we use the value of the left-most truncated bit to determine if rounding should occur. This scheme is almost the same as the IEEE 754 scheme, but without the special case for values halfway between two representations. The results showed a slightly worse error than with the full IEEE rounding scheme. If we use simple bit truncation, with no explicit rounding (equivalent to rounding towards zero always), we find that the model suffers from numerical instability. Similar results were found during the course of previous studies with different numerical simulations (Russell et al., 2015). It seems that rounding towards infinity would give similar results to always rounding towards zero, likely inducing numerical instabilities. We have added some further detail to the text on this topic (page: 9; line: 27):

> "The shallow water model provides a good context to discuss the effect of rounding on conservation properties. The surface height integrated over the model domain is conserved in the shallow water equations. However in a numerical approximation we expect that there will be some error introduced due to truncation. For the reference double precision simulation the mean surface height of the fluid changes by $\sim 10^{-17}$ m over 20 million time-steps, in the single precision simulation the change is $\sim 10^{-7}$ m, and in the co-designed half precision simulation it is $\sim 10^{-3}$ m. Although the height error at lower precision is increased greatly over the double precision simulation, the absolute amount of fluid lost is small compared to the domain size. If this is a concern then a mass-fixing scheme could be introduced to mitigate. Perhaps more interesting is the influence of the rounding scheme used when truncating floating-point significands. If we truncate without rounding (equivalent to always rounding towards zero), a large bias is introduced into the simulation, to the point where it will fail to complete due to numerical errors. If instead we choose to round to the nearest representation (as IEEE but without the special case for numbers half way between two representations illustrated in Fig. 3d) we see a height error of $\sim 10^{-2}$ . The difference between the round-to-nearest and the round-to-nearest with tie-to-even rounding schemes is not negligible, highlighting the value of unbiased rounding on computing conserved properties."

As a remark, being not familiar with fluid dynamics simulations, I cannot easily juge the similarity of results displayed in Figure 4. The similarity of results displayed in Figure 5 seems clearer. However it would be interesting to compare them with those obtained with 10-bit significand.

We have included a more complete description of the features in Fig. 4, which we hope will better allow readers who are not familiar with fluid dynamics to better assess the performance for themselves (page: 9; line: 4):

> "The simulation using single precision (Fig. 4b) is qualitatively similar to the reference double-precision simulation (Fig. 4a) in that it shows height variations with similar magnitudes, with no suggestion of numerical stability problems. These snapshots are not expected to be identical between the different precision configurations due to the chaotic nature of the simulations, but similar flow behaviour would be expected. The simulation with emulated 15-bit significands (Fig. 4c) also shows smooth fields with height features of similar magnitude as

*in the double-precision reference simulation. The surface height field in the simulation with 10-bit significands (Fig. 4d) has smaller magnitude variations than the reference simulation, and some small-scale noise features can be seen as jagged contours, particularly noticeable in the lower part of the domain. These features suggest the dynamics of the simulation with 10-bit significands are being perturbed by a lack of precision."*

We chose not to include a panel for the 10-bit significand simulation in Fig. 5, since we have already established the 10-bit significand simulation is probably not appropriate by examining the instantaneous flow in Fig. 4. We have produced a version of this figure including the 10-bit significand model version for this response (AR-Fig. 1). This shows that although the time-mean circulation pattern is qualitatively reasonable, there is a weaker height gradient along the western boundary in the 10-bit significand model (AR-Fig. 1c) than we see in the other 3 models, which confirms that this model is not performing as well as the higher precision or the co-designed models.

The performance overhead due to the library is mentioned in the conclusion. It would be interesting to have measurements of that overhead.

We have included extra text in the Discussion and conclusions section estimating the performance overhead of *rpe* (page: 11; line: 3):

*"The overhead introduced by the emulator may vary depending on its application. In our experience the overhead of the library in terms of execution speed is at least a factor of 10. Tests on the simple chaotic dynamical system of Lorenz (1963) suggest an execution time penalty of approximately 30 times, compared to a double precision simulation, and the co-designed shallow water model detailed in Sect. 3 experiences an execution time penalty of approximately 70 times compared to the double precision implementation, both tested using gfortran 4.8 with optimization level 2 to compile both the models and rpe itself."*

Technical corrections are proposed below.

p4: The reduced-precision program (Fig. 1b) set –> The reduced-precision program (Fig. 1b) sets

Done.

p7: Therefore we generally we discourage –> Therefore we generally discourage

Done.

[revised manuscript text omitted]